# PISA: Compressive Sensing Adaptation of Large Language Models

## Abstract

In this paper, we introduce a novel perspective on Parameter-Efficient Fine-Tuning (PEFT) by viewing the weight update matrix as a k-sparse approximation in the spatial domain, departing from the commonly used low-rank structure assumption. We propose a compressive sensing-based approach that leverages under-complete measurement matrices to analyze the approximation capabilities of the weight update matrix. Our method ensures bounded error in the reconstruction of the weight updates, as guaranteed by theoretical results in compressive sensing. However, the vectorization of the weight update matrix leads to a high-dimensional problem ($d^2$), which can potentially result in large error bounds. To address this issue, we introduce a block-structured approximation scheme that partitions the weight update matrix into smaller blocks and applies the $k$-sparse approximation to each block independently. We theoretically analyze the approximation error bounds of our approach and demonstrate that the block-structured scheme achieves tighter error bounds compared to the non-block approach. Empirically, we validate the effectiveness of our proposed method on various downstream NLP tasks, showcasing its ability to achieve competitive performance with a reduced number of trainable parameters. Our approach offers a new direction for parameter-efficient fine-tuning of large language models. Notably, our experiments demonstrate competitive performance with only 500 learnable parameters, while offering greater memory and computational efficiency than LoRA in a rank-1 setting.

## 1 Introduction

The rapid advancement of large pre-trained language models has revolutionized natural language processing (NLP) tasks. Models such as BERT (Devlin et al., 2019), GPT (Radford et al., 2018), and T5 (Raffel et al., 2020b) have attained remarkable performance in a wide range of downstream tasks. However, the scaling up of large foundation models has led to soaring costs in fine-tuning and parameter storage, rendering extensive adaptations impractical. For instance, regular full fine-tuning of a LLaMA2-7B parameter model(Chen et al., 2023) requires more than 60GB of GPU memory, which exceeds the capacity of common consumer GPUs(Pan et al., 2024). This challenge has sparked the development of Parameter-Efficient Fine-Tuning (PEFT) techniques(Houlsby et al., 2019). They adapt large pre-trained language models to downstream tasks by only fine-tuning a small number of (extra) model parameters, which simultaneously diminishing the quantity of trainable parameters and retaining high-level performance (Ding et al., 2023).

Among PEFT techniques, Low-Rank Adaptation (LoRA) (Hu et al., 2022) has gained significant attention due to its effectiveness and efficiency. Comparable or better downstream performances has been observed on various NLP tasks, including text classification, question answering, and language generation (Mao et al., 2024). LoRA adapts pre-trained models by introducing low-rank update matrices to the model's weights. Specifically, LoRA represents the weight update matrix as the product of two low-rank matrices which are learned during fine-tuning, while the pre-trained weights remain frozen. Although LoRA achieves parameter efficiency by introducing pluggable low-rank matrices. As these LoRA plugins accumulate, the computation cost of is increasing and unignorable. It is necessary to further enhance the computation efficiency of LoRA. Ongoing efforts have been made to further improve the computational efficiency of LoRA from the perspectives of parameter freezing, parameter pruning, and parameter sharing (Wu et al., 2024; Zhang et al., 2023a; Liu et al., 2024b; Bałazy et al., 2024; Zhou et al., 2024).

In the domain of Parameter-Efficient Fine-Tuning (PEFT) for large language models, low-rank adaptations have been the prevailing approach. However, the limitations of low-rank structures in capturing complex patterns have led us to a crucial insight: weight updates in transformers can often be represented very sparsely in an appropriate basis. This observation naturally points us towards compressive sensing, a paradigm uniquely suited to exploit such sparsity. We introduce comPressIve Sensing Adaption (`PISA`), a method that reimagines weight updates as highly sparse signals in a high-dimensional space, offering unprecedented flexibility and efficiency in model adaptation. While this shift to high-dimensionality initially seems to present a challenge – as the Restricted Isometry Property (RIP) typically predicts larger error bounds in such spaces – the extreme sparsity of weight updates allows us to overcome this hurdle. To fully leverage this sparsity, we propose a novel block-structured k-sparse approximation scheme. This approach partitions the weight update matrix into manageable blocks, enabling efficient computation and tighter error bounds. By further exploiting properties of the Fourier transform, such as Hermitian symmetry and $\ell_1$ minimization as projections, we enhance parameter efficiency and effectiveness even more. Our comprehensive theoretical analysis provides rigorous error bounds and computational complexity assessments, while extensive empirical validation demonstrates `PISA`'s competitive performance on various NLP tasks with significantly fewer parameters than existing methods. The main contributions of this paper are as follows:

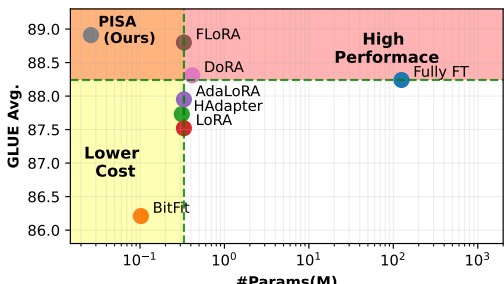

Figure 1: Efficiency *vs.* effectiveness on the GLUE dataset with RoBert-Large. Our `PISA` enjoys high average and uses fewer params. than competitors.

1. We propose a block-structured k-sparse approximation scheme that partitions the weight update matrix into smaller blocks, allowing for more efficient computation and tighter error bounds. This approach, combined with properties of the Fourier transform such as Hermitian symmetry, significantly enhances parameter efficiency.

2. We provide a comprehensive theoretical analysis of `PISA`, including error bounds and computational complexity. Our analysis reveals how the interplay between sparsity, block size, and measurement matrix properties affects the adaptation quality and computational efficiency.

3. Through extensive empirical validation on various downstream NLP tasks, we demonstrate that `PISA` achieves competitive performance with significantly fewer trainable parameters compared to existing PEFT methods. Our experiments showcase the practical viability of our approach in adapting large language models efficiently.

## 2 RELATED WORKS

**Large Language Models.** Large language models, such as BERT (Devlin et al., 2019), RoBERTa (Liu, 2019), GPT (Radford et al., 2018), and LLaMA (Touvron et al., 2023a);(Touvron et al., 2023b);(inc, 2024) have achieved remarkable success in various natural language processing tasks. The swift expansion of the parameter scales of pre-training language models considerably improves their generalization performance and gives rise to the emergence of novel capabilities. Nevertheless, the capabilities of large language models (LLMs) on some downstream tasks remains limited due to their inherent knowledge boundaries. To expand the knowledge boundaries, it is necessary to fine-tune LLMs on the downstream tasks. Over the past few years, the parameter scales of pre-training language models have witnessed a several-thousand-fold increase. For instance, it has gone from a model with 330 million parameters like BERT (Devlin et al., 2019) to one with 540 billion parameters such as PaLM (Chowdhery et al., 2022). This characteristic renders fine-tuning them for specific tasks is extremely computationally expensive and resource-intensive.

**Parameter-Efficient Fine-Tuning.** Fine-tuning large language models for downstream tasks typically involves updating all the model parameters using task-specific training data. Nevertheless, as the model size escalates, this approach becomes impractical due to the substantial computa-

tional resources and storage overhead it demands. To reduce the computational cost, numerous parameter-efficient fine-tuning (PEFT) methods have been proposed (Ding et al., 2023). These methods enable large language models (LLMs) to adapt to downstream tasks by fine-tuning only a small number of (additional) model parameters.One notable approach is the LoRA (Low-Rank Adaptation) method (Hu et al., 2022), which introduces low-rank matrices to approximate the weight updates during fine-tuning. LoRA factorizes the weight updates into low-rank matrices, significantly reducing the number of trainable parameters. However, LoRA still necessitates storing the low-rank matrices, which can be memory-consuming for large models. Other parameter-efficientfine-tuning methods comprise adapter-based approaches (Houlsby et al., 2019),which introduce small trainable modules between the layers of the pre-trained model, and prefix-tuning (Li & Liang, 2021), which prepends trainable continuous prompts to the input sequences. Although these methods have demonstrated promising outcomes in reducing the number of trainable parameters, they still encounter challenges in terms of memory efficiency and the capacity to capture complex weight update patterns.

**Compressive Sensing** Compressive sensing has emerged as a powerful framework for efficient signal acquisition and reconstruction, with applications spanning various fields including signal processing, imaging, and machine learning (Candès et al., 2006; Donoho, 2006). The foundational work of Candes et al. (2006) and Donoho (Donoho, 2006) established theoretical guarantees for recovering sparse signals from a small number of linear measurements. These guarantees rely on properties such as the Restricted Isometry Property (RIP) (Candes & Tao, 2005), which has been shown to hold for various classes of measurement matrices, including random Gaussian matrices and partial Fourier matrices (Rudelson & Vershynin, 2008). Simultaneously, in the domain of parameter-efficient fine-tuning, methods like LoRA (Hu et al., 2021) have garnered popularity due to their capability to adapt large pre-trained models with a small number of parameters. LoRA accomplishes this by learning low-rank update matrices, where the number of parameters is determined by the selected rank. Our work bridges these two areas, applying compressive sensing principles to parameter-efficient fine-tuning of large language models. Unlike traditional compressive sensing approaches, we concentrate on approximating weight update matrices rather than input signals. In contrast to LoRA,the parameter efficiency of our method is based on the chosen number of parameters rather than a rank value, thereby offering enhanced flexibility.

## 3 PRELIMINARIES

In this section, we introduce the notation and formally define the problem of parameter-efficient fine-tuning for large language models. We begin by discussing the general form of weight updates and then present the LoRA method as a specific instantiation. Finally, we introduce the concept of basis representations, which serves as a foundation for our proposed `PISA` method.

**Notation** Let $\mathbf{W} \in \mathbb{R}^{d \times d}$ be the weight matrix of a pre-trained language model, where $d$ is the dimension of the model's hidden states. During fine-tuning, we aim to learn an update matrix $\Delta \mathbf{W} \in \mathbb{R}^{d \times d}$ that adapts the pre-trained weights to a specific downstream task. The updated weight matrix $\mathbf{W}'$ can be expressed as:

$$\mathbf{W}' = \mathbf{W} + \Delta \mathbf{W} \tag{1}$$

The adapted weights W' are then used in the linear layer to compute the output y:

$$\boldsymbol{y} = \boldsymbol{W}'\boldsymbol{x} = (\boldsymbol{W} + \Delta \boldsymbol{W})\boldsymbol{x} \tag{2}$$

**Reparameterized Fine-tuning.** The weight update matrix $\Delta W$ acts as a filter that modifies the linear transformation performed by the pre-trained weights W. The filter $\Delta W$ learns to emphasize or suppress certain patterns or features in the input x that are relevant to the downstream task. We can represent the filter $\Delta W$ using a basis set A and a learnable representation B:

$$\Delta \boldsymbol{W} = \boldsymbol{A}\boldsymbol{B} \tag{3}$$

The choice of the basis set A determines the type of patterns or structures that the filter $\Delta \boldsymbol{W}$ can learn to capture. In LoRA, both $\boldsymbol{A}$ and $\boldsymbol{B}$ are learned simultaneously, allowing the filter $\Delta \boldsymbol{W}$ to capture task-specific patterns and structures. The low-rank structure of $\boldsymbol{A}$ and $\boldsymbol{B}$ reduces the number of trainable parameters. In LoRA-FA (Zhang et al., 2023a), only $\boldsymbol{B}$ is learned, while $\boldsymbol{A}$ remains fixed. This approach further reduces the number of trainable parameters compared to LoRA, as the basis set is predetermined. The fixed basis set $\boldsymbol{A}$ captures general patterns or structures relevant to the task.

**Compressive Sensing for PEFT.** Compressive sensing is a signal processing technique that enables the reconstruction of a sparse signal from a small number of linear measurements (Candès & Wakin, 2008). We propose that $\Delta\mathbf{W}$ can be approximated using compressed measurements (Donoho, 2006), leveraging principles from compressive sensing. This approach is motivated by the following key result:

**Lemma 1 (Approximation from Compressed Measurements (Baraniuk et al., 2008))** *Let* $\mathbf{x}$ *be an $N$-dimensional vector, and let* $\mathbf{x}_k$ *be its best $k$-sparse approximation. Let* $\mathbf{A}$ *be an $r \times d$ matrix satisfying certain properties, where* $r \geq C \cdot (k\log(d/k) + \log(1/\delta))/\varepsilon^2$ *for some constants $C > 0$ and $0 < \delta < 1$. Then, with probability at least $1 - \delta$, the error of approximating $\mathbf{x}$ using the matrix $\mathbf{A}$ satisfies:*

$$\|\mathbf{x} - \mathbf{A}^\dagger\mathbf{A}\mathbf{x}\|_2 \leq \|\mathbf{x} - \mathbf{x}_k\|_2 + \varepsilon\|\mathbf{x}\|_2 \tag{4}$$

*where* $\mathbf{A}^\dagger$ *denotes the Moore-Penrose pseudoinverse of* $\mathbf{A}$.

This lemma provides a theoretical foundation for our approach, suggesting that we can achieve accurate reconstruction of $\Delta\mathbf{W}$ from compressed measurements, provided that $\Delta\mathbf{W}$ is well-approximated by a sparse matrix. The error bound consists of two terms: the approximation error of the best $k$-sparse representation and a term proportional to $\varepsilon$, which can be controlled by the number of measurements.

# 4 METHODOLOGY

In this section, we reframe Parameter-Efficient Fine-Tuning (PEFT) within the context of compressive sensing, viewing the weight update matrix as a k-sparse signal in a high-dimensional space. This novel perspective allows us to address the parameter inefficiency problem inherent in full fine-tuning by leveraging the power of sparse representations. Specifically, we propose to approximate the weight update matrix using a compressed sensing framework, where a small number of measurements can capture the essen-

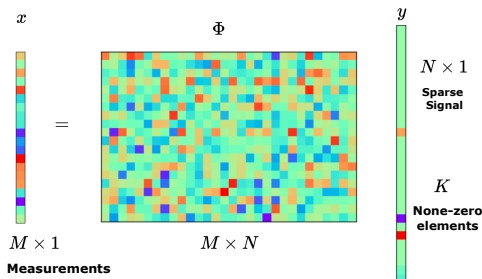

Figure 2: Comp. sensing Adaption (k-sparse).

tial information of the sparse update. However, this high-dimensional approach initially faces the challenge of potentially large error bounds, as predicted by the Restricted Isometry Property (RIP). To overcome this, we introduce a block-structured k-sparse approximation scheme, which partitions the weight update matrix into smaller, more manageable blocks. This approach not only allows for more efficient computation but also leads to tighter error bounds. We provide a comprehensive theoretical analysis of our method, utilizing the RIP to derive approximation error bounds and demonstrate the effectiveness of our block-structured approach. Through this analysis, we establish the conditions under which our method can achieve accurate approximations with significantly fewer parameters than traditional fine-tuning or other PEFT techniques.

## 4.1 PROBLEM DEFINITION AND COMPRESSIVE SENSING FRAMEWORK

We propose a novel perspective that views $\Delta\mathbf{W}$ through the lens of compressive sensing. Instead of assuming a low-rank structure, we posit that $\Delta\mathbf{W}$ can be approximated as a $k$-sparse matrix in the spatial domain. To leverage compressive sensing techniques, we first vectorize $\Delta\mathbf{W}$ as $\text{vec}(\Delta\mathbf{W}) \in \mathbb{R}^{d^2}$. Our compressive sensing problem for weight updates can then be formulated as:

$$\boldsymbol{b} = \mathbf{A}\text{vec}(\Delta\mathbf{W}) \tag{5}$$

where $\mathbf{A} \in \mathbb{R}^{m \times d^2}$ is a measurement matrix, and $\boldsymbol{b} \in \mathbb{R}^m$ represents $m$ linear measurements of $\text{vec}(\Delta\mathbf{W})$. Crucially, we work with under-complete measurements, meaning $m < d^2$. This formulation allows us to potentially capture more complex update patterns than the low-rank approximation, while still maintaining parameter efficiency. However, it also introduces two significant challenges: **Memory Inefficiency** and **High Dimensionality**. The measurement matrix $\mathbf{A}$, despite being fixed during training, requires storing $m \times d^2$ parameters. For large language models where $d$ can be in the thousands, this becomes prohibitively memory-intensive. The vectorization of $\Delta\mathbf{W}$ results

in a very high-dimensional vector ($d^2$), which can lead to large error bounds in the compressive sensing reconstruction (Lemma 1). In the following subsections, we introduce novel techniques to address these challenges, enabling the practical application of compressive sensing principles to parameter-efficient fine-tuning of large language models.

## 4.2 Addressing Parameters Inefficiency

To tackle the challenge of memory inefficiency in storing the large measurement matrix $\mathbf{A}$, we introduce two key concepts: the Kronecker product in conjunction with the Restricted Isometry Property (RIP), and the use of subsampled Fast Fourier Transform (FFT) matrices.

**Kronecker Product and RIP.** The Restricted Isometry Property (RIP) is a fundamental concept in compressive sensing that guarantees the stability of sparse signal recovery. A matrix $\boldsymbol{\Phi} \in \mathbb{R}^{m \times n}$ satisfies the RIP of order $k$ with constant $\delta_k \in (0, 1)$ if, for all $k$-sparse vectors $\mathbf{x} \in \mathbb{R}^n$:

$$(1 - \delta_k)\|\mathbf{x}\|_2^2 \leq \|\boldsymbol{\Phi}\mathbf{x}\|_2^2 \leq (1 + \delta_k)\|\mathbf{x}\|_2^2 \tag{6}$$

To extend this property to our two-dimensional weight update matrix, we leverage the Kronecker product. Let $\boldsymbol{\Phi}_1 \in \mathbb{R}^{m_1 \times d}$ and $\boldsymbol{\Phi}_2 \in \mathbb{R}^{m_2 \times d}$ be matrices satisfying the RIP. We can construct our measurement matrix $\mathbf{A}$ as:

$$\mathbf{A} = \boldsymbol{\Phi}_1 \otimes \boldsymbol{\Phi}_2 \tag{7}$$

where $\otimes$ denotes the Kronecker product. Importantly, if $\boldsymbol{\Phi}_1$ and $\boldsymbol{\Phi}_2$ satisfy the RIP of order $k$ with constants $\delta_1$ and $\delta_2$ respectively, then $\mathbf{A}$ satisfies the RIP of order $k^2$ with constant $\delta_1 + \delta_2 + \delta_1\delta_2$ (Duarte & Baraniuk, 2010; 2011). This construction allows us to work with structured measurement matrices, reducing the storage requirements from $O(md^2)$ to $O(m_1d + m_2d)$. However, while this is a significant improvement, it still requires storing parameters for the measurement matrix.

**Subsampled FFT Matrix.** To further reduce memory requirements while maintaining the Kronecker product structure, we propose using subsampled Fast Fourier Transform (FFT) matrices (Xu & Xu, 2014) as our measurement matrices $\boldsymbol{\Phi}_1$ and $\boldsymbol{\Phi}_2$. This approach allows us to efficiently transform from the spatial domain to the spectral (k-sparse) domain. An additional crucial reason for using subsampled FFT matrices is their orthogonality property. In the context of LLM fine-tuning, traditional $\ell_1$ minimization approaches (solving $A\Delta W = b$) become computationally intractable. However, with orthogonal matrices like subsampled FFT, we can directly compute $\delta W = A^H b$, where $\mathbf{A}^H$ is the Hermitian transpose of $\mathbf{A}$. This simple computation replaces the need for complex L1 minimization, making our approach feasible and efficient for LLM fine-tuning. Let $\mathbf{F}_d \in \mathbb{C}^{d \times d}$ denote the 1D discrete Fourier transform (DFT) matrix. We construct our measurement matrices as:

$$\boldsymbol{\Phi}_1 = \mathbf{S}_1\mathbf{F}_d, \quad \boldsymbol{\Phi}_2 = \mathbf{S}_2\mathbf{F}_d \tag{8}$$

where $\mathbf{S}_1 \in \mathbb{R}^{m_1 \times d}$ and $\mathbf{S}_2 \in \mathbb{R}^{m_2 \times d}$ are random subsampling matrices. Each row of $\mathbf{S}_1$ and $\mathbf{S}_2$ contains a single 1 in a random position and 0s elsewhere. Our complete measurement matrix $\mathbf{A}$ is then constructed as:

$$\mathbf{A} = \boldsymbol{\Phi}_1 \otimes \boldsymbol{\Phi}_2 = (\mathbf{S}_1\mathbf{F}_d) \otimes (\mathbf{S}_2\mathbf{F}_d) \tag{9}$$

This construction maintains the Kronecker product structure while leveraging the properties of the Fourier transform. Importantly, subsampled FFT matrices have been shown to satisfy the RIP with high probability (Haviv & Regev, 2017), making them suitable for our compressive sensing framework. The use of subsampled FFT matrices in this Kronecker product framework offers **Extreme Memory Efficiency** and **Computational Efficiency**. We only need to store the k-sparse matrix $\mathbf{B}$ in the frequency domain, which requires $O(k)$ memory, where $k \ll d^2$. We do not need to store any part of the measurement matrix $\mathbf{A}$. The FFT operation can be implemented very efficiently, with a time complexity of $O(d^2 \log d)$ for a $d \times d$ matrix. The measurement process can be expressed as:

$$b = \mathbf{A}^H \text{vec}(\Delta\mathbf{W}) = \text{vec}(\mathbf{S}_1(\mathbf{F}_d^H \Delta\mathbf{W}\mathbf{F}_d)\mathbf{S}_2^T) \tag{10}$$

where $\mathbf{F}_d^H$ is the Hermitian transpose of $\mathbf{F}_d$. This formulation allows us to efficiently compute the measurements using 2D FFT operations followed by subsampling. By combining the Kronecker product structure with subsampled FFT matrices, we effectively address the memory inefficiency challenge while maintaining the theoretical guarantees provided by the RIP. This approach allows us to apply compressive sensing techniques to large-scale neural network fine-tuning in a memory-efficient and computationally effective manner, leveraging the transition from spatial to spectral domains.

4.3 TACKLING HIGH DIMENSIONALITY AND ERROR BOUNDS

While our FFT-based approach significantly reduces memory requirements, we still face challenges due to the high dimensionality of the weight update matrix $\Delta\mathbf{W} \in \mathbb{R}^{d\times d}$. High dimensionality can lead to large error bounds in the compressive sensing reconstruction process. To address this, we introduce two key techniques: exploiting Hermitian symmetry and implementing a block-structured k-sparse approximation.

**Exploiting Hermitian Symmetry.** An important property of the Discrete Fourier Transform (DFT) of real-valued signals is Hermitian symmetry. For our real-valued weight update matrix $\Delta\mathbf{W}$, its 2D Fourier transform $\mathcal{F}(\Delta\mathbf{W})$ exhibits the following symmetry:

$$\mathcal{F}(\Delta\mathbf{W})[u,v] = \mathcal{F}(\Delta\mathbf{W})^*[-u,-v] \tag{11}$$

where $^*$ denotes the complex conjugate. This symmetry allows us to further reduce the number of parameters we need to learn and store. Specifically, we only need to learn and store roughly half of the frequency components in $\mathbf{B}$ while the other half can be reconstructed using the Hermitian symmetry property. This reduction in parameters not only improves memory efficiency but also potentially reduces the error in the sparse approximation by effectively doubling the sparsity level for the same number of learned parameters.

**Block-Structured k-Sparse Approximation.** To further address the challenges posed by high dimensionality, we introduce a block-structured k-sparse approximation. Instead of treating $\Delta\mathbf{W}$ as a single large matrix, we partition it into smaller blocks:

$$\Delta\mathbf{W} = \begin{bmatrix} \Delta\mathbf{W}_{11} & \Delta\mathbf{W}_{12} & \cdots & \Delta\mathbf{W}_{1n} \\ \Delta\mathbf{W}_{21} & \Delta\mathbf{W}_{22} & \cdots & \Delta\mathbf{W}_{2n} \\ \vdots & \vdots & \ddots & \vdots \\ \Delta\mathbf{W}_{n1} & \Delta\mathbf{W}_{n2} & \cdots & \Delta\mathbf{W}_{nn} \end{bmatrix} \tag{12}$$

where each $\Delta\mathbf{W}_{ij} \in \mathbb{R}^{b\times b}$, and $b = d/n$ is the block size. For each block, we apply our FFT-based k-sparse approximation independently:

$$\mathbf{b}_{ij} = \text{vec}(\mathbf{S}_{1,ij}(\mathbf{F}_b^H \Delta\mathbf{W}_{ij}\mathbf{F}_b)\mathbf{S}_{2,ij}^T) \tag{13}$$

where $\mathbf{S}_{*,ij}$ is a block-specific subsampling operator. By working with smaller blocks, we reduce the effective dimensionality of each compressive sensing problem. This leads to tighter error bounds (in the next section)for the reconstruction of each block. By combining the exploitation of Hermitian symmetry with the block-structured k-sparse approximation, we effectively address the challenges posed by the high dimensionality of $\Delta\mathbf{W}$. This approach not only leads to tighter error bounds but also offers greater flexibility and computational efficiency in the fine-tuning process.

4.4 THEORETICAL ANALYSIS

In this subsection, we provide a theoretical analysis of our block-structured k-sparse approximation approach, focusing on the approximation bounds and expressive power. We leverage results from compressed sensing theory, particularly the Restricted Isometry Property (RIP), to derive upper bounds on the approximation error.

**Theorem 1 (Effect of Block Size on Approximation Error)** *Let $vec(\Delta\mathbf{W}) \in \mathbb{R}^{N^2}$ be the vectorized form of the weight update matrix, and let $(vec(\Delta\mathbf{W}))_k$ be its best k-sparse approximation in the frequency domain. Consider partitioning $vec(\Delta\mathbf{W})$ into $B_1$ blocks of size $N^2/B_1$ and $B_2$ blocks of size $N^2/B_2$, where $B_1 < B_2$. Let $\mathbf{A}_i^{(1)} \in \mathbb{C}^{m_1\times(N^2/B_1)}$ and $\mathbf{A}_i^{(2)} \in \mathbb{C}^{m_2\times(N^2/B_2)}$ be subsampled FFT matrices for the i-th block in each partitioning scheme, satisfying the RIP of order k with constant $\epsilon$, where $m_1 \geq C \cdot (k\log(N^2/(B_1 k)) + \log(1/\delta))/\epsilon^2$ and $m_2 \geq C \cdot (k\log(N^2/(B_2 k)) + \log(1/\delta))/\epsilon^2$ for some constants $C > 0$ and $0 < \delta < 1$. Then, with*

*probability at least $1 - \delta$, the average approximation errors for the two block sizes satisfy:*

$$\frac{1}{B_2} \sum_{i=1}^{B_2} \|vec(\Delta \mathbf{W})_i^{(2)} - \mathbf{A}_i^{(2)\dagger} \mathbf{A}_i^{(2)} vec(\Delta \mathbf{W})_i^{(2)}\|_2$$

$$\leq \frac{1}{B_1} \sum_{i=1}^{B_1} \|vec(\Delta \mathbf{W})_i^{(1)} - \mathbf{A}_i^{(1)\dagger} \mathbf{A}_i^{(1)} vec(\Delta \mathbf{W})_i^{(1)}\|_2$$

$$\leq \|vec(\Delta \mathbf{W}) - (vec(\Delta \mathbf{W}))_k\|_2 + \epsilon \|vec(\Delta \mathbf{W})\|_2,$$

*where $vec(\Delta \mathbf{W})_i^{(l)}$ denotes the $i$-th block of $vec(\Delta \mathbf{W})$ in the $l$-th partitioning scheme, and $\mathbf{A}_i^{(l)\dagger}$ is the pseudoinverse of $\mathbf{A}_i^{(l)}$.*

This theorem demonstrates that, under the same conditions (i.e., the same sparsity level $k$ and RIP constant $\epsilon$), using a larger number of smaller blocks results in a tighter approximation error bound. The intuition behind this result is that using smaller blocks allows for better adaptation to local structures within the vectorized weight update, leading to a more accurate approximation in the frequency domain.

**Theorem 2 (Expressive Power of K-Sparse Compressive Sensing)** *Let $\bar{f}(\mathbf{x}) = \bar{\mathbf{W}}\mathbf{x}$ be the target linear model and $f_0(\mathbf{x}) = \prod_{l=1}^{L} \mathbf{W}_l \mathbf{x}$ be the pre-trained linear model, where $\bar{\mathbf{W}}, \mathbf{W}_l \in \mathbb{R}^{d \times d}$ for $l \in [L]$. Define the error matrix $\mathbf{E} = \bar{\mathbf{W}} - \prod_{l=1}^{L} \mathbf{W}_l$. Let $\mathbf{A} \in \mathbb{R}^{m \times d^2}$ be a measurement matrix satisfying the RIP of order $k$ with constant $\delta_k \in (0, 1)$, where $m < d^2$. For each layer $l \in [L]$, let $\mathbf{b}_l \in \mathbb{R}^m$ be a vector such that $\Delta \mathbf{W}_l = \text{vec}^{-1}(\mathbf{A}^\dagger \mathbf{b}_l)$, where $\mathbf{A}^\dagger$ is the pseudoinverse of $\mathbf{A}$ and $\text{vec}^{-1}$ is the inverse vectorization operation. Assume that all weight matrices of the frozen model $(\mathbf{W}_l)_{l=1}^{L}$, and $\prod_{l=1}^{L} \mathbf{W}_l + \text{vec}^{-1}(\mathbf{A}^\dagger \mathbf{b}_l)$ are non-singular for any $\mathbf{b}_l \in \mathbb{R}^m$. Then, for the adapted model $f(\mathbf{x}) = \prod_{l=1}^{L} (\mathbf{W}_l + \Delta \mathbf{W}_l)\mathbf{x}$, we have:*

$$\left\| \text{vec}\left( \prod_{l=1}^{L} (\mathbf{W}_l + \Delta \mathbf{W}_l) - \bar{\mathbf{W}} \right) \right\|_2 \leq \|\text{vec}(\mathbf{E}) - \mathbf{A}^\dagger \mathbf{A} \text{vec}(\mathbf{E})\|_2 + \frac{\delta_k}{\sqrt{1 - \delta_k}} \|\mathbf{A}^\dagger \mathbf{A} \text{vec}(\mathbf{E})\|_2, \quad (14)$$

*Moreover, if $\mathbf{A}$ has full row rank, there exist $\mathbf{b}_l \in \mathbb{R}^m$ for $l \in [L]$ such that $\prod_{l=1}^{L} (\mathbf{W}_l + \Delta \mathbf{W}_l) = \bar{\mathbf{W}}$, implying $f = \bar{f}$.*

This theorem demonstrates that, under the compressive sensing framework, the approximation error bound is influenced by the structure of the measurement matrix $\mathbf{A}$ and the number of layers $L$ in the model. Specifically, increasing the number of layers $L$ while keeping the total number of parameters (determined by $m$) constant effectively results in using a larger number of smaller "blocks" to approximate the weight update.

### 4.5 COMPUTATIONAL COMPLEXITY ANALYSIS

To fully appreciate the efficiency of our proposed method, it's crucial to analyze its computational complexity in comparison to traditional approaches like full fine-tuning and LoRA. We consider both time and space complexity for forward and backward phases. Compared to LoRA, our method can

Table 1: Complexity Comparison of Fine-Tuning Methods

| Method | Forward Pass | Backward Pass | Space |
|---|---|---|---|
| Full Fine-tuning | $O(d^2)$ | $O(d^2)$ | $O(d^2)$ |
| LoRA | $O(d^2 + 2dr)$ | $O(d^2 + 2dr)$ | $O(2dr)$ |
| PISA | $O(d^2 + (d/b)^2 \cdot (k + b^2 \log b))$ | $O(d^2 + (d/b)^2 \cdot (k + b^2 \log b))$ | $O((d/b)^2 \cdot k)$ |

be more efficient when $k \log b < r$, which is often the case for large models with a small number of significant frequency components. Our method's space complexity can be lower than LoRA when $k < rb/d$. This condition is often met in practice, especially for large models where we can maintain a high degree of sparsity. In summary, our method offers a favorable trade-off between

Table 2: Results on GLUE for natural language understanding tasks. We report the overall (matched and mismatched) accuracy for MNLI, Matthew's correlation for CoLA, Pearson correlation for STS-B, and accuracy for other tasks. Higher is better for all metrics. We also report the number of trainable parameters (#Params) for each method.

| | Method | #Params | MNLI | SST-2 | CoLA | QQP | QNLI | RTE | MRPC | STS-B | Avg |
|---|---|---|---|---|---|---|---|---|---|---|---|
| | Fully FT | 1000‰ | 87.62 | 94.84 | 63.58 | **91.87** | 92.80 | 78.80 | 90.20 | **91.23** | 86.37 |
| RoB-Base | IA | 0.44‰ | 84.83 | 94.15 | 60.14 | 87.92 | 90.39 | 76.17 | 87.75 | 90.23 | 83.91 |
| | SSL | 0.22‰ | 83.45 | 93.81 | 56.02 | 87.30 | 89.20 | 74.01 | 86.76 | 89.52 | 82.51 |
| | SSB | 0.66‰ | 85.80 | 94.61 | 60.92 | 88.65 | 91.20 | 76.53 | 86.76 | 90.23 | 84.34 |
| | BitFit | 0.82‰ | 85.29 | 94.61 | 59.58 | 88.10 | 91.20 | 79.78 | 88.73 | 90.32 | 84.70 |
| | HAdapter | 2.50‰ | 87.45 | 94.72 | 63.88 | 90.29 | 92.71 | 80.14 | **89.22** | 90.80 | 86.15 |
| | PAdapter | 2.43‰ | 87.11 | 94.15 | 62.74 | 89.95 | 92.71 | 80.14 | 87.99 | 90.13 | 85.62 |
| | LoRA | 2.65‰ | 87.20 | 94.38 | 65.61 | 89.25 | 92.07 | 81.59 | 87.99 | 91.01 | 86.14 |
| | TriLoRA | 2.65‰ | 86.81 | 94.61 | 64.47 | 89.61 | 91.82 | 76.53 | 88.24 | 90.31 | 85.30 |
| | AdaLoRA | 2.65‰ | 87.31 | 94.72 | 64.33 | 89.77 | 92.81 | 81.95 | 88.24 | 90.48 | 86.20 |
| | FLoRA | 2.65‰ | 87.31 | 94.38 | 64.09 | 89.97 | 92.77 | **82.67** | 87.75 | 90.77 | 86.21 |
| | DoRA | 3.32‰ | 86.74 | 94.50 | 66.19 | 90.28 | 91.95 | 79.78 | 88.48 | 91.01 | 86.12 |
| | PISA | 0.22‰ | **87.69** | **95.08** | **66.80** | 89.99 | **92.80** | 79.88 | 88.90 | 91.10 | **86.53** |
| RoB-Large | Fully FT | 1000‰ | 89.90 | 95.63 | 69.19 | **92.40** | 94.03 | 83.75 | 89.46 | 91.60 | 88.24 |
| | BitFit | 0.76‰ | 89.37 | 94.84 | 66.96 | 88.41 | 92.24 | 78.80 | 87.75 | 91.35 | 86.21 |
| | HAdapter | 2.35‰ | 90.10 | 95.41 | 67.65 | 91.19 | 93.52 | 83.39 | 89.25 | 91.31 | 87.73 |
| | PAdapter | 2.29‰ | 89.89 | 94.72 | 69.06 | 91.05 | 93.87 | 84.48 | 89.71 | 91.38 | 88.02 |
| | LoRA | 2.49‰ | 90.03 | 93.92 | 69.15 | 90.61 | 93.37 | 85.56 | 90.19 | 90.75 | 87.95 |
| | TriLoRA | 2.49‰ | 90.22 | 95.77 | 69.51 | 91.18 | 94.15 | 87.00 | **91.07** | 91.38 | 88.89 |
| | AdaLoRA | 2.49‰ | 90.40 | 95.80 | 69.98 | 91.43 | 94.23 | 87.36 | 90.43 | **91.63** | 88.90 |
| | FLoRA | 2.49‰ | 90.60 | 96.00 | 70.20 | 90.40 | 94.46 | **88.81** | 90.93 | 90.56 | 88.80 |
| | DoRA | 3.12‰ | 90.21 | 94.38 | 69.33 | 90.84 | 93.26 | 86.94 | 90.19 | 91.34 | 88.31 |
| | PISA | 0.16‰ | **90.91** | **96.10** | **70.32** | 90.84 | **94.76** | 86.74 | 90.59 | 91.24 | **88.91** |

computational complexity and parameter efficiency, especially for large models. It provides a scalable approach to fine-tuning that can be more efficient than both full fine-tuning and low-rank methods like LoRA, particularly when the weight updates can be well-approximated by sparse frequency domain representations.

## 5 EXPERIMENTS

**Setting.** We set `block_size = 16` and `m = 2000` for `PISA` and rank of LoRA to $r = 8$ as default. To ensure a fair comparison, we initially fine-tuned models with `PISA` following the LoRA configuration *e.g.*, weight initialization, learning rate, *etc.*, and maintained the same training steps for both `PISA` and LoRA when fine-tuning on the same datasets. We conduct experiment on three tasks including GLUE benchmark, commonsense reasoning, and MMLU. The codebases for baselines implementation and evaluation are sourced from their official GitHub repositories/library (*i.e.*, Vision Task, GLUE, and MMLU are from Gao et al. (2024), Si et al. (2024), and Zheng et al. (2024), respectively).

### 5.1 GLUE BENCHMARK

In GLUE experiments, we employed one small scales of transformer, *RoBERTa-base* (Liu, 2019), as the base model. We used the General Language Understanding Evaluation (GLUE) (Raffel et al., 2020a) benchmark as our dataset, which comprises two single-sentence classification tasks, three similarity and paraphrase tasks, and four natural language inference tasks. There are two prominent series of extension-based methods within parameter-efficient tuning. The first series, the Adapter derivatives, comprises methods such as those introduced by Houlsby et al. (2019), Houlsby et al. (2019), and introduced by Pfeiffer et al. (2020); Zaken et al. (2021), which incorporate small-scale neural modules, or adapters, into existing architectures. The second series, known as LoRA derivatives, includes developments such as LoRA (Hu et al., 2021), AdaLoRA (Zhang et al., 2023b), TriLoRA (Feng et al., 2024), FLoRA (Hao et al., 2024), DoRA (Liu et al., 2024a), and DyLoRA (Valipour et al., 2023), AdaLoRA (Zhang et al., 2023b), These methods primarily rely on low-rank matrix decomposition techniques.

Table 3: MMLU scores for `PISA` and other PEFT methods, showcasing `PISA`'s ability to achieve high performance while maintaining parameter efficiency across base models. Best performance is indicated by the bold face numbers.

|  | FT-Method | # Params | STEM | Social | Human | Other | Average |
|---|---|---|---|---|---|---|---|
| LLaMA3-8B | FT | 1000‰ | 52.93 | 73.40 | 59.06 | 69.34 | 63.26 |
|  | LoRA | 7.00‰ | **54.45** | 74.82 | 58.96 | 70.23 | 64.10 |
|  | PISA | 0.30‰ | 54.43 | **74.90** | **60.56** | **70.84** | **64.77** |
| Mistral-7B | FT | 1000‰ | 50.00 | 68.07 | 53.12 | 65.01 | 58.09 |
|  | LoRA | 8.30‰ | **50.60** | 68.87 | 53.62 | 65.21 | 58.99 |
|  | PISA | 0.34‰ | 50.50 | **68.97** | **53.92** | **65.28** | **59.12** |
| LLaMA2-13B | FT | 1000‰ | 46.23 | 64.47 | 49.34 | 61.23 | 55.31 |
|  | LoRA | 6.70‰ | **46.56** | 64.77 | 49.67 | 61.76 | 55.69 |
|  | PISA | 0.28‰ | 46.44 | **64.85** | **49.89** | **61.80** | **55.75** |

Table 4: Comparison of image classification accuracy (%) on various datasets using ViT-Base and ViT-Large models with Linear Probing (LP), LoRA, and PISA fine-tuning methods. Results show performance across seven diverse image datasets, with `n=3k/6k` samples and `block_size = 16` for PISA. Best performance for each dataset and model size is highlighted in bold

|  | Method | OxfordPets | StanfordCars | CIFAR10 | EuroSAT | FGVC | RESISC45 | CIFAR100 | Avg. |
|---|---|---|---|---|---|---|---|---|---|
| ViT-Base | LP | 90.28 | 25.76 | 96.41 | 88.72 | 17.44 | 74.22 | 84.28 | 68.16 |
|  | LoRA | 93.19 | 45.38 | **98.78** | 98.44 | 25.16 | 92.70 | 92.02 | 77.95 |
|  | PISA | **95.33** | **53.28** | 98.20 | **98.64** | **31.86** | **94.07** | **98.44** | **81.40** |
| ViT-Large | LP | 91.11 | 37.9 | 97.78 | 92.64 | 24.62 | 82.02 | 84.28 | 72.91 |
|  | LoRA | 94.82 | **73.25** | 99.13 | 98.65 | 39.92 | 93.86 | 93.31 | 84.71 |
|  | PISA | **97.45** | 69.67 | **99.35** | **98.92** | **43.63** | **94.24** | **93.61** | **85.27** |

Table 2 presents the results from experiments on the GLUE benchmark. Note that `PISA` achieves competitive performance while using significantly fewer trainable parameters compared to full fine-tuning and otherLoRA variants (*e.g.*, TriLoRA, AdaLoRA) support the argument that moving beyond low-rank structures can be beneficial for model adaptation. This aligns with the our goal of developing more parameter-efficient methods for adapting large language models. The compressive sensing-based approaches like `PISA` can offer a new direction for parameter-efficient fine-tuning of large language models, providing flexibility and efficiency advantages over existing methods.

## 5.2 INSTRUCTION TUNING

We evaluate the downstream task performance of `PISA`. We utlize three language models *LLaMA3-8B* (inc, 2024), *Mistral-7B* (Jiang et al., 2023), and *LLaMA2-13B* (Touvron et al., 2023a). we employ the instruction-following finetuning task with Alpaca GPT-4(en) dataset, which consists instances generated by GPT-4 (OpenAI, 2023) based on inputs from Alpaca (Taori et al., 2023). We adopt the **The Massive Multitask Language Understanding benchmark (MMLU)** (Hendrycks et al., 2020) to test our model. It consists of multiple-choice questions sourced from various fields, including humanities, social sciences, and STEM.

Table 3 shows that PISA's performance is consistently strong across different model sizes and architectures (LLaMA3-8B, Mistral-7B, LLaMA2-13B), demonstrating its versatility.Despite using significantly fewer parameters, PISA achieves comparable or better performance than both full fine-tuning and LoRA across different models and categories.

## 5.3 VISION TASKS.

We conduct the evaluation of our method on the image classification task. We employ the Base and Large versions of the popular CV foundation model, Vision Transformer (ViT) (Dosovitskiy, 2020). The ViTs are pretrained on the ImageNet-21K dataset (Ridnik et al., 2021). The datasets for fine-tuning include OxfordPets $(37^2)$, CIFAR10 (10), DTD (47), EuroSAT (10) and RESISC45 (45) with small label spaces, as well as StanfordCars (196), FGVC (100) and CIFAR100 (100) with large label spaces.

(a) Different `m` in PISA results, sst2-base dataset, `block_size=16`

| m | #Param | Accuracy |
|---|---|---|
| 500 | 0.05‰ | 92.16 |
| 1000 | 0.11‰ | 93.43 |
| 2000 | 0.22‰ | 94.46 |
| 4000 | 0.44‰ | **94.58** |

(b) different `block_size` in PISA results, sst2-base dataset, `m=2000`

| block_size | #Param | Accuracy |
|---|---|---|
| 4 | 0.22‰ | 92.43 |
| 8 | 0.22‰ | 93.66 |
| 16 | 0.22‰ | 94.46 |
| 32 | 0.22‰ | **94.80** |

Table 4 presents results for image classification tasks using Vision Transformer (ViT) models. PISA consistently outperforms Linear Probing (LP) across all datasets and both model sizes. It also shows competitive or superior performance compared to LoRA in most cases. PISA's strong performance is maintained across both ViT-Base and ViT-Large models, showing its effectiveness for different model sizes.

## 5.4 Ablation on Hyperparameters

To understand the influence of key hyperparameters on our model's performance, we conducted ablation studies on two critical factors: the number of measurements (m) and the block size. Tables 5b and 5a present the results of these studies, showing how different values affect the model's accuracy on the sst2-base dataset. Increasing m generally improves accuracy, with diminishing returns beyond $m = 2000$. The most significant improvement occurs between m=500 and m=2000. Larger block sizes consistently yield better accuracy. The improvement is more pronounced for smaller block sizes, with diminishing returns as block size increases.

## 6 Conclusion

In this paper, we have introduced a novel approach to Parameter-Efficient Fine-Tuning (PEFT) that leverages the principles of compressive sensing. By viewing the weight update matrix as a compressed representation in the measurement domain, we depart from the conventional low-rank structure assumptions prevalent in current PEFT methods. Our theoretical analysis, supported by empirical evidence, demonstrates that this approach can effectively adapt pre-trained models to new tasks while significantly reducing the number of trainable parameters. The key innovation of our method lies in its ability to capture complex adaptation patterns without the constraints of low-rank structures. This flexibility, combined with the efficiency of working directly in the compressed domain, addresses several limitations of existing techniques such as LoRA and its variants. Notably, our approach maintains a constant computational overhead regardless of adaptation complexity, a significant advantage over methods where computational costs increase with accumulated adaptations. Our theoretical framework provides clear bounds on the approximation error, offering insights into the trade-offs between parameter efficiency and adaptation quality. This rigorous foundation not only enhances our understanding of PEFT but also guides practical implementations. The scalability of our method becomes particularly apparent as model sizes grow, with compressive measurements scaling sub-linearly with model size, making it a sustainable approach for adapting very large models. The empirical validation of our method across various downstream NLP tasks underscores its practical viability. By achieving competitive performance with a reduced number of trainable parameters, we demonstrate that theoretical elegance can translate into tangible benefits in real-world scenarios.

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

## A  PROOF

### A.1  PROOF OF THEOREM 1

**Proof 1** *We will prove this theorem in several steps:*

***Step 1: RIP Property for Individual Blocks*** *For each block $i$ in partitioning scheme $l$ (where $l \in \{1, 2\}$), the subsampled FFT matrix $\mathbf{A}_i^{(l)}$ satisfies the RIP of order $k$ with constant $\epsilon$. This means that for any $k$-sparse vector $\mathbf{x}$ of appropriate dimension:*

$$(1 - \epsilon)\|\mathbf{x}\|_2^2 \leq \|\mathbf{A}_i^{(l)}\mathbf{x}\|_2^2 \leq (1 + \epsilon)\|\mathbf{x}\|_2^2 \tag{15}$$

***Step 2: Approximation Error for a Single Block*** *Let $\mathbf{x}_i^{(l)}$ be the $i$-th block of $vec(\Delta\mathbf{W})$ in the $l$-th partitioning scheme, and let $(\mathbf{x}_i^{(l)})_k$ be its best $k$-sparse approximation. From the RIP property and known results in compressed sensing (Candes et al., 2006), we have:*

$$\|\mathbf{x}_i^{(l)} - \mathbf{A}_i^{(l)\dagger}\mathbf{A}_i^{(l)}\mathbf{x}_i^{(l)}\|_2 \leq \|\mathbf{x}_i^{(l)} - (\mathbf{x}_i^{(l)})_k\|_2 + \epsilon\|\mathbf{x}_i^{(l)}\|_2 \tag{16}$$

***Step 3: Average Error Bound for Partitioning Scheme*** *Taking the average over all blocks in partitioning scheme $l$:*

$$\frac{1}{B_l}\sum_{i=1}^{B_l}\|\mathbf{x}_i^{(l)} - \mathbf{A}_i^{(l)\dagger}\mathbf{A}_i^{(l)}\mathbf{x}_i^{(l)}\|_2 \leq \frac{1}{B_l}\sum_{i=1}^{B_l}\|\mathbf{x}_i^{(l)} - (\mathbf{x}_i^{(l)})_k\|_2 + \epsilon\frac{1}{B_l}\sum_{i=1}^{B_l}\|\mathbf{x}_i^{(l)}\|_2 \tag{17}$$

***Step 4: Relating Block-wise and Global Approximations*** *Let $(vec(\Delta\mathbf{W}))_k$ be the best $k$-sparse approximation of the entire vector. We can show that:*

$$\frac{1}{B_l}\sum_{i=1}^{B_l}\|\mathbf{x}_i^{(l)} - (\mathbf{x}_i^{(l)})_k\|_2 \leq \|vec(\Delta\mathbf{W}) - (vec(\Delta\mathbf{W}))_k\|_2 \tag{18}$$

*This inequality holds because the left-hand side represents a potentially sub-optimal $k$-sparse approximation (block-wise) compared to the global optimal on the right-hand side.*

***Step 5: Applying Cauchy-Schwarz Inequality*** *Using the Cauchy-Schwarz inequality:*

$$\frac{1}{B_l}\sum_{i=1}^{B_l}\|\mathbf{x}_i^{(l)}\|_2 \leq \sqrt{\frac{1}{B_l}\sum_{i=1}^{B_l}\|\mathbf{x}_i^{(l)}\|_2^2} = \frac{1}{\sqrt{B_l}}\|vec(\Delta\mathbf{W})\|_2 \tag{19}$$

***Step 6: Combining Results*** *Substituting the results from Steps 4 and 5 into the inequality from Step 3:*

$$\frac{1}{B_l}\sum_{i=1}^{B_l}\|\mathbf{x}_i^{(l)} - \mathbf{A}_i^{(l)\dagger}\mathbf{A}_i^{(l)}\mathbf{x}_i^{(l)}\|_2 \leq \|vec(\Delta\mathbf{W}) - (vec(\Delta\mathbf{W}))_k\|_2 + \frac{\epsilon}{\sqrt{B_l}}\|vec(\Delta\mathbf{W})\|_2 \tag{20}$$

***Step 7: Comparing Block Sizes*** *Since $B_2 > B_1$, we have:*

$$\frac{\epsilon}{\sqrt{B_2}} < \frac{\epsilon}{\sqrt{B_1}} \tag{21}$$

*Therefore:*

$$\begin{aligned}
&\frac{1}{B_2}\sum_{i=1}^{B_2}\|\mathbf{x}_i^{(2)} - \mathbf{A}_i^{(2)\dagger}\mathbf{A}_i^{(2)}\mathbf{x}_i^{(2)}\|_2 \\
&\leq \frac{1}{B_1}\sum_{i=1}^{B_1}\|\mathbf{x}_i^{(1)} - \mathbf{A}_i^{(1)\dagger}\mathbf{A}_i^{(1)}\mathbf{x}_i^{(1)}\|_2 \\
&\leq \|vec(\Delta\mathbf{W}) - (vec(\Delta\mathbf{W}))_k\|_2 + \epsilon\|vec(\Delta\mathbf{W})\|_2
\end{aligned} \tag{22}$$

*This completes the proof of the theorem.*

## A.2 Proof of Theorem 2

**Proof 2** *Let $\mathbf{E} = \bar{\mathbf{W}} - \prod_{l=1}^{L} \mathbf{W}_l$ be the error matrix. First, we express the adapted model in terms of the error matrix:*

$$\prod_{l=1}^{L}(\mathbf{W}_l + \Delta\mathbf{W}_l) = \prod_{l=1}^{L}\mathbf{W}_l + \sum_{l=1}^{L}\left(\prod_{i=l+1}^{L}\mathbf{W}_i\right)\Delta\mathbf{W}_l\left(\prod_{j=1}^{l-1}(\mathbf{W}_j + \Delta\mathbf{W}_j)\right) + \textit{higher order terms}$$
(23)

$$= \bar{\mathbf{W}} - \mathbf{E} + \sum_{l=1}^{L}\Delta\mathbf{W}_l + \textit{higher order terms},$$
(24)

*where we've used the fact that $\prod_{l=1}^{L}\mathbf{W}_l = \bar{\mathbf{W}} - \mathbf{E}$.*

*Now, let's consider the vectorized form:*

$$\text{vec}\left(\prod_{l=1}^{L}(\mathbf{W}_l + \Delta\mathbf{W}_l) - \bar{\mathbf{W}}\right) = -\text{vec}(\mathbf{E}) + \sum_{l=1}^{L}\text{vec}(\Delta\mathbf{W}_l) + \textit{higher order terms}$$
(25)

*Recall that $\Delta\mathbf{W}_l = \text{vec}^{-1}(\mathbf{A}^\dagger\mathbf{b}_l)$. Substituting this in:*

$$\text{vec}\left(\prod_{l=1}^{L}(\mathbf{W}_l + \Delta\mathbf{W}_l) - \bar{\mathbf{W}}\right) = -\text{vec}(\mathbf{E}) + \sum_{l=1}^{L}\mathbf{A}^\dagger\mathbf{b}_l + \textit{higher order terms}$$
(26)

*We can choose $\mathbf{b}_l$ such that $\sum_{l=1}^{L}\mathbf{b}_l = \mathbf{A}\text{vec}(\mathbf{E})$. This gives:*

$$\text{vec}\left(\prod_{l=1}^{L}(\mathbf{W}_l + \Delta\mathbf{W}_l) - \bar{\mathbf{W}}\right) = -\text{vec}(\mathbf{E}) + \mathbf{A}^\dagger\mathbf{A}\text{vec}(\mathbf{E}) + \textit{higher order terms}$$
(27)

*Taking the $\ell_2$ norm of both sides and using the triangle inequality:*

$$\left\|\text{vec}\left(\prod_{l=1}^{L}(\mathbf{W}_l + \Delta\mathbf{W}_l) - \bar{\mathbf{W}}\right)\right\|_2 \leq \|\text{vec}(\mathbf{E}) - \mathbf{A}^\dagger\mathbf{A}\text{vec}(\mathbf{E})\|_2 + \|\textit{higher order terms}\|_2$$
(28)

*Now, let's focus on bounding $\|\text{vec}(\mathbf{E}) - \mathbf{A}^\dagger\mathbf{A}\text{vec}(\mathbf{E})\|_2$. We can write:*

$$\|\text{vec}(\mathbf{E}) - \mathbf{A}^\dagger\mathbf{A}\text{vec}(\mathbf{E})\|_2^2 = \|\text{vec}(\mathbf{E})\|_2^2 - \|\mathbf{A}\text{vec}(\mathbf{E})\|_2^2$$
(29)

*This is because $(\mathbf{I} - \mathbf{A}^\dagger\mathbf{A})$ and $\mathbf{A}^\dagger\mathbf{A}$ are orthogonal projections. Using the RIP property of $\mathbf{A}$, we have:*

$$(1 - \delta_k)\|\text{vec}(\mathbf{E})\|_2^2 \leq \|\mathbf{A}\text{vec}(\mathbf{E})\|_2^2 \leq (1 + \delta_k)\|\text{vec}(\mathbf{E})\|_2^2$$
(30)

*Substituting this into the previous equation:*

$$\|\text{vec}(\mathbf{E}) - \mathbf{A}^\dagger\mathbf{A}\text{vec}(\mathbf{E})\|_2^2 \leq \delta_k\|\text{vec}(\mathbf{E})\|_2^2$$
(31)

*Taking the square root:*

$$\|\text{vec}(\mathbf{E}) - \mathbf{A}^\dagger\mathbf{A}\text{vec}(\mathbf{E})\|_2 \leq \sqrt{\delta_k}\|\text{vec}(\mathbf{E})\|_2$$
(32)

*Now, $\|\mathbf{A}^\dagger\mathbf{A}\text{vec}(\mathbf{E})\|_2 \leq \frac{1}{\sqrt{1-\delta_k}}\|\text{vec}(\mathbf{E})\|_2$ by the RIP property. Combining these results and neglecting higher order terms for the upper bound:*

$$\left\|\text{vec}\left(\prod_{l=1}^{L}(\mathbf{W}_l + \Delta\mathbf{W}_l) - \bar{\mathbf{W}}\right)\right\|_2 \leq \sqrt{\delta_k}\|\text{vec}(\mathbf{E})\|_2 + \frac{\delta_k}{\sqrt{1-\delta_k}}\|\text{vec}(\mathbf{E})\|_2$$
(33)

*For the exact representation case, if $\mathbf{A}$ has full row rank, then $\mathbf{A}^\dagger\mathbf{A}$ is the identity when restricted to the row space of $\mathbf{A}$. Thus, we can choose $\mathbf{b}_l$ such that $\sum_{l=1}^{L}\mathbf{A}^\dagger\mathbf{b}_l = \text{vec}(\mathbf{E})$, allowing for exact representation. This completes the proof.*

