# OpenReview forum: "PISA: Compressive Sensing Adaptation of Large Language Models"
_ICLR.cc/2025/Conference — Submitted to ICLR 2025_

### Official Review · Reviewer_FxtU · 2024-10-31

**Soundness:** 2
**Presentation:** 3
**Contribution:** 2
**Rating:** 3
**Confidence:** 3

**Summary:**

The paper introduces a parameter-efficient compression method for fine-tuning LLM based on principles from compressed sensing. The core idea is to assume that the weight matrix update during fine-tuning is sparse, enabling lossless compression via a Restricted Isometry Property (RIP) matrix into a shorter vector, thereby reducing memory usage.

**Strengths:**

Building on the idea of Compressed Sensing, the authors propose further enhancements through matrix subdivision, use of the Kronecker product, and the subsampled FFT matrix to improve performance. The reported numerical results are impressive, showing substantial memory reduction compared to baseline methods. The paper is well-written.

**Weaknesses:**

**Theoretical Gaps:** The theoretical foundation presented in this paper has several gaps that impact the proposed approach’s overall reliability. Please see the specific concerns with Lemma 1 below for details.

**Inference Time Consideration:** While memory efficiency is certainly valuable, inference time is often an even more critical factor. A slight increase in memory usage would be acceptable if it resulted in reduced inference times. Currently, however, the paper lacks any comparison of computation times.

**Concerns with Lemma 1 (Key Theoretical Motivation)**
 -The authors indicate that Lemma 1 is an established result, yet I was unable to locate it in the cited references.
 - It is unusual for a lemma to refer to "certain conditions" without specifying these conditions explicitly. If space constraints are a concern, please consider including the full theorem in the appendix.
 - In compressed sensing, using the Moore-Penrose pseudoinverse for signal reconstruction is atypical, as it generally results in suboptimal performance. Classical compressed sensing papers typically present reconstruction guarantees for basis pursuit (constrained ℓ1-norm minimization), and the cited reference similarly provides results only for basis pursuit. It is unclear to me how these guarantees apply to reconstruction using the Moore-Penrose pseudoinverse.
 - Lemma 1 includes a probability of failure δ, yet all parameters are defined deterministically. Could the authors please clarify the source of randomness here?
 - Most importantly, Lemma 1 is formulated for a single, fixed signal rather than for all signals in an open set. As I understand the result, inequality (4) does not hold universally for all signals in the search space. Since the proposed approach involves optimization over signals (i.e., weights) during training, the inequality would need to hold for all signals in this space, which this lemma does not guarantee. Thus, Lemma 1 may not be strong enough to substantiate the proposed approach.

**Questions:**

Could the authors please indicate the exact location of the source for Lemma 1 in the references? Also, I suggest adding comparisons of inference times (e.g., by comparing the number of FLOPs of the compressed network). The FLOP count could be computed using available Python packages.

I would consider raising the score if the concerns are adequately addressed.

---

### Official Review · Reviewer_4tRc · 2024-10-31

**Soundness:** 3
**Presentation:** 2
**Contribution:** 2
**Rating:** 5
**Confidence:** 4

**Summary:**

should generally agree with a well-written summary.
The authors propose to exploit a compressive sensing framework to train LLM adaptors. The proposed method allows to train a few parameters while being comparative to other LoRa-stye methods. A theoretical analysis of used blocks size and approximation error bound is provided. The method is evaluated on text and visual benchmarks.

**Strengths:**

- The idea of using compressive sensing for LLM adapters  is novel and interesting
- The work includes both theoretical contributions and methodology-based empirical contributions.
- The method allows to train very small number of parameters while provides comparative accuracy results.

**Weaknesses:**

**Major**:

- efficiency evaluation of the method is missing to support the claim of the authors: “*Our experiments showcase the practical viability of our approach in adapting large language models efficiently*” in the introduction.

- the motivation is not convincing, especially in lines 49-52: “*As these LoRA plugins accumulate, the computation cost of is increasing and unignorable*”. LoRa plugins are very small in number of parameters already compared to the LLM base model.

- I think that the theorems should be backed up with empirical analysis that demonstrate the main results

- The concluding step after Eq 13 - once we can compute b, how do we optimize $\Delta W$ from Eq. 2?

- there are missing more recent baselines (for RoBERTA backbones), e.g.: LoRA-XS, LoRA-FA, VeRA [1-3]

- Missing parameters numbers in Table 4.




**Minor**:
- Lines 49-52 - the sentences should be combined and rephrased: “Although LoRA… .As these LoRA plugins accumulate..”
- Please define 2D FFT before providing the property in Eq 11
- The title of Table 5 is missing
- Instead of presenting absolute numbers for $block_size$, I think it is better to present it also as rates relative to the original matrix dimension (16 out of 1024).


**References**:

[1] Zhang, Longteng, et al. "Lora-fa: Memory-efficient low-rank adaptation for large language models fine-tuning." arXiv preprint arXiv:2308.03303 (2023).

[2] Bałazy, Klaudia, et al. "LoRA-XS: Low-Rank Adaptation with Extremely Small Number of Parameters." arXiv preprint arXiv:2405.17604 (2024).

[3] Kopiczko, Dawid J., Tijmen Blankevoort, and Yuki M. Asano. "Vera: Vector-based random matrix adaptation." arXiv preprint arXiv:2310.11454 (2023).

**Questions:**

- Section 4.5 - I think you claim about efficiency is not correct ,the complexity you have is  $O(d^2 + (d/b)^2 + d^2logb) > O(d^2 + 2dr) $ for any d. Pease check it again.

- I’m not sure I understand why do you use Hermitian transpose on A in Eq. 10 while it is not used in the Eq. 5?

- The transition to the right side in Eq. 10 in unclear, please specify intermediate steps and explain how it is obtained if the matrix-vector product is defined by $(A \otimes B ) vec(V) = vec(BVA^T)$

- What matrices do you update, attention matrices only? Which of them?

---

### Official Review · Reviewer_R7zz · 2024-11-01

**Soundness:** 1
**Presentation:** 1
**Contribution:** 2
**Rating:** 3
**Confidence:** 4

**Summary:**

The authors tackle the challenge of efficiently adapting a large language model to a new task. This challenge, known as parameter-efficient fine-tuning, has gained significant attention in recent years, leading to the development of multiple methods. This work introduces a new perspective on the problem by using compressed sensing. The core idea is to represent the adaptor matrix in the frequency domain, thereby reducing the number of parameters needed for updates. Since the compressed sensing formulation leads to a high-dimensional problem, they proposed using blocks of the adaptor matrix as a sparse approximation to reduce memory and computational complexity. The new method is evaluated on several datasets, demonstrating that it is competitive compared to existing schemes.

**Strengths:**

The paper addresses an important problem.

The compressed sensing perspective to the problem is new.

The authors provide an analysis of the effect of block size and the expressive power of the K-sparse compressed sensing representation.

**Weaknesses:**

The paper is not well written, with multiple mistakes and typos.

Several important details are not clearly explained in the paper. Specifically, some steps required for implementing the method are convolved within the theory without a clear explanation for the reader of the steps required for updating the weights in practice.

There are some mistakes in the computational and memory complexity analysis. Those are not followed by any practical example that demonstrates that the method is, in fact, beneficial in these aspects. The authors demonstrate that they only update a few parameters, but since those should be translated back to the weight space, it is unclear whether there is computational or memory gain (compared to Lora or Bitfit, for example).

**Questions:**

In practice the procedure for updating $\Delta W$ is not explained, are you using $vec^{-1}(A^{\dagger}b)$? Why doesn’t this step appear in the complexity analysis in Table 1?

There are multiple mistakes and typos in the paper:

Missing space before citations in lines 38,39,40.

The caption of figure 1: do you mean average accuracy? Also, there is a . in the middle of the second sentence.

Missing space in lines 111 and 116.

Wrong (and inconsistent) citation format in line 124 Candes vs. Donoho.

Notation is very messy, with many instances of abuse of notation (A and B mentioned multiple times but play different roles) and instances of math symbols not being represented as math symbols. For example, line 155 A and B should be math. Line 153 W.
Line 153 $\Delta W $ should be bold. The list goes on.

Capital B is used again later for a different context without clearly explaining its relation to b.

Dimensions of multiple terms were not explained, for example, A and B on page 3.

The term measurement matrix on page 4 is standard in compressed sensing but could be confusing in the context of supervised learning since it means something completely different.

What is the relation between m_1, m_2, and m? (page 5)

L1 minimization written in two ways in the same paragraph (page 5)

You mention that the FFT is very efficient (page 5), but this requires d^2logd operations; this seems larger than any term mentioned in Table 1. Therefore, how can this be considered efficient? And why doesn’t this term appear as part of your complexity in Table 1?

The complexity in Table 1 does not coincide with the conclusions; specifically, comparing memory space to Lora leads to the following result:

$k<2rb^2/d$ for PISA to be more space efficient than Lora. In contrast to what is written in the paper.

In terms of forward and backward pass, I fail to understand how you conclude the complexity of the method is lower than that of Lora for the mentioned condition.

Also, while the authors mention that the method is more efficient than other methods in practice, there isn’t any practical example that demonstrates this. Can you compare the FULL memory and computational runtime requirements to Lora and Bitftit?

Wrong citation format in section 5. It should be with brackets.

In section 5.4, $m$ is inconsistent and should be a math symbol.

---

### Official Review · Reviewer_mB2w · 2024-11-04

**Soundness:** 2
**Presentation:** 2
**Contribution:** 3
**Rating:** 3
**Confidence:** 2

**Summary:**

This paper propose PISA, a compressive sensing based LoRA variant.
PISA treats learning weights updates as finding k-sparse approximation of the optimal updates in some spatial domain.
Several tweaks to reduce error bound are also applied.
The proposed method achieved competitive performance on both vision and language models compared to representative baselines.

**Strengths:**

1. The problem studied in this paper is important.
2. The experiment design looks sufficient to me.

**Weaknesses:**

While the direction explored in this paper can be valuable, the draft in its current form need significant improvement from several aspects:

1. The motivation of the proposed compressive sensing perspective is not well-stated. In line 58 the authors stated that "*the limitations of low-rank
structures in capturing complex patterns have
led us to a crucial insight: weight updates
in transformers can often be represented very
sparsely in an appropriate basis*",
which is confusing to me. The low-rank structure limitation is an inherent one, while the sparse weight updates should be largely empirical. I don't see how the limitation of LoRA directly motivates the compressive sensing view.
In other words, what new insight this compressive sensing view brings is not clearly stated.
In my opinion, more convincing and intuitive explanation of why PISA is conceptually better than existing LoRA variants is critical.

2. The proposed method is hard to comprehend as well. After reading the method section, I am still confused in what sense the *sparsity* of PISA is about. In addition, the authors did not mention in PISA what are learnable parameters and how they are learned. The authors should be more explicit and provide more intuitive explanation of the proposed method as well. An algorithm block is highly encouraged.

3. In addition, the technical novelty, at least in this current presentation form, looks limited to me. For now the key innovative part is the introduction of compressive sensing view to LoRA. But this part, as mentioned above, is not well motivated. The main body of the present Method is using existing compressive sensing technique to reduce computation load. The connection between the theoretical analysis and LLM fine-tuning is weak, in particular, the authors didn't clearly state what assumptions are made about LLM parameters, as a result, I am unconvinced about their validity.

**Questions:**

1. According to Table 4, why isn't #params affected by block_size?

See above questions in Weakness.

---

### Meta-Review · Area_Chair_xPqL · 2024-12-23

**Metareview:**

This paper introduces PISA, a parameter-efficient fine-tuning method for large language models based on compressive sensing principles. The authors view weight updates as k-sparse approximations in the spatial domain and propose a block-structured scheme to address high-dimensional problems. They provide theoretical error bounds and evaluate the method on various NLP tasks, claiming competitive performance with only 500 learnable parameters.

The paper's main strength is its application of compressive sensing to parameter-efficient fine-tuning, along with theoretical analysis and competitive performance on various tasks. However, significant weaknesses undermine its potential impact. The motivation and intuition behind the compressive sensing approach are not clearly articulated, and technical details crucial for implementation are often unclear or missing. The theoretical foundations, particularly Lemma 1, have notable gaps. Efficiency claims lack sufficient empirical support, and comparisons to some recent relevant baselines are absent.

The decision to reject is based on these key issues: inadequate explanation of the conceptual advantages of the compressive sensing approach, lack of clarity in practical implementation, insufficient empirical evidence for efficiency claims, gaps in the theoretical foundation, and incomplete comparisons with recent baselines. These concerns significantly impact the paper's clarity, soundness, and reproducibility, requiring substantial revisions beyond the scope of a simple rebuttal.

**Additional Comments On Reviewer Discussion:**

No rebuttal was submitted.

---

### Decision · Program_Chairs · 2025-01-22

Reject